# VDAC in Retinal Health and Disease

**DOI:** 10.3390/biom14060654

**Published:** 2024-06-04

**Authors:** Ying Xu, Shanti R. Tummala, Xiongmin Chen, Noga Vardi

**Affiliations:** 1Guangdong Key Laboratory of Non-Human Primate Research, Key Laboratory of CNS Regeneration (Ministry of Education), Guangdong-Hongkong-Macau Institute of CNS Regeneration, Jinan University, Guangzhou 510632, China; xuying@jnu.edu.cn (Y.X.); chenxm@jnu.edu.cn (X.C.); 2Department of Pharmacology, University of Pennsylvania, Philadelphia, PA 19104, USA; stummala@pennmedicine.upenn.edu; 3Department of Neuroscience, University of Pennsylvania, Philadelphia, PA 19104, USA

**Keywords:** photoreceptors, ganglion cells, HK2, mitochondria, retinitis pigmentosa, age related macular degeneration, glaucoma, apoptosis, oxidative stress, degeneration

## Abstract

The retina, a tissue of the central nervous system, is vital for vision as its photoreceptors capture light and transform it into electrical signals, which are further processed before they are sent to the brain to be interpreted as images. The retina is unique in that it is continuously exposed to light and has the highest metabolic rate and demand for energy amongst all the tissues in the body. Consequently, the retina is very susceptible to oxidative stress. VDAC, a pore in the outer membrane of mitochondria, shuttles metabolites between mitochondria and the cytosol and normally protects cells from oxidative damage, but when a cell’s integrity is greatly compromised it initiates cell death. There are three isoforms of VDAC, and existing evidence indicates that all three are expressed in the retina. However, their precise localization and function in each cell type is unknown. It appears that most retinal cells express substantial amounts of VDAC2 and VDAC3, presumably to protect them from oxidative stress. Photoreceptors express VDAC2, HK2, and PKM2—key proteins in the Warburg pathway that also protect these cells. Consistent with its role in initiating cell death, VDAC is overexpressed in the retinal degenerative diseases retinitis pigmentosa, age related macular degeneration (AMD), and glaucoma. Treatment with antioxidants or inhibiting VDAC oligomerization reduced its expression and improved cell survival. Thus, VDAC may be a promising therapeutic candidate for the treatment of these diseases.

## 1. Retinal Structure and Function

The voltage-dependent anion channel (VDAC), most commonly present in the outer mitochondrial membrane, shuttles metabolites between the mitochondrion and cytosol, and plays a key role in apoptosis (programmed cell death). As will be apparent from the following description, the unique characteristics of the retina raise several important questions regarding the function of VDAC in retinal cells. The purpose of this review is to summarize what is known about VDAC isoforms in the retina and to speculate about their proposed functions.

The retina’s function is to capture light from objects in the world, process it, and transmit the resulting image information to the brain. In most vertebrates, light is captured by two classes of photoreceptors: rods, which are highly sensitive and capture dim light, and several types of cones that function mainly in bright light and are responsible for color vision. The information captured by photoreceptors is transmitted to second-order cells—the bipolar cells, which come in two main flavors, the ON cells that mainly code light intensities that are higher than the background, and the OFF cells which code light intensities lower than the background. The information from bipolar cells is further processed and sent to the brain by the third-order cells—the ON, OFF, and ON-OFF retinal ganglion cells (RGCs). Two additional cell classes are involved in the information processing that improve contrast and feature detection. Horizontal cells in the outer plexiform layer collect information from neighboring photoreceptors and feed it to bipolar cells and amacrine cells in the inner plexiform layer collect information from neighboring bipolar cells and feed it to ganglion cells (Figure 1A; for further information, see detailed reviews [1,2,3,4]).

In addition to neurons, the retina contains several types of glial cells and is supported by the retinal pigment epithelium (RPE) and the retinal and choroidal circulation (see reviews [5,6]). The pigment epithelium is a mono layer of cells in between the outer segments of the photoreceptors and the choroid plexus, a vascular tissue that serves as the energy source for the photoreceptors. The RPE forms the outer blood retinal barrier, and it nourishes the photoreceptors, phagocytoses their outer segments, and recycles the rod opsin 11-Cis retinal. The glial cells include Muller cells, astrocytes, and microglia (see review [7]). Muller cells span the cross-section of the retina, and their multiple thin processes envelop most neurons. Their main function is to isolate the neurons so that their potential is retained, to recycle the cone opsins, to supply nutrients to the cells they envelope, and to ensure neuronal response sensitivity by taking up released neurotransmitters. Astrocytes reside in the nerve fiber layer and couple inner retinal neuronal function with blood flow in the retinal circulation. The microglia, resident macrophages of the central nervous system, are situated in the outer and inner plexiform layers, ready to act when an infectious agent or injury is detected.

Several aspects of retinal structure and function make it unique within the nervous system. (1) Photoreceptors are the most metabolically active cells in the body. Their high metabolism is due to at least two functional designs. (a) Their resting membrane potential. Unlike most neurons whose resting voltage potential across the membrane is around −70 mV, photoreceptors ‘rest’ in darkness at about −35 mV, a voltage that is achieved by opening the cGMP-gated channels on the plasma membrane [8]. (b) The daily phagocytosis and renewal of their outer segment discs [9]. Approximately a tenth of the outermost discs are phagocytosed by the pigment epithelial cells and replaced by newly synthesized discs in the inner region of the outer segment. This constant renewal requires a continuous production of polyunsaturated fatty acids and proteins. (2) The retina produces a high level of reactive oxygen species (ROS), a consequence of the fast metabolism and continuous light exposure [10,11,12,13]. The retina is designed such that the light sensors (the photoreceptors) are at the back of the eye, supported and maintained by the pigment epithelium and choroid. While this ensures an optimal performance of the sensors, it exposes the entire neural retina to photo (free radical) damage. Consequently, several specialized metabolic processes have evolved to protect the individual components from oxidative damage and ensure functionality. In particular, the mitochondria in the photoreceptors have diverse and dynamic structures.

## 2. Mitochondria in Photoreceptors Are Unique and Dynamic

### 2.1. Structure

In most neurons, including most retinal cells and RPE, mitochondria are present in the cell body near the nucleus. In photoreceptors, the mitochondria are not in their soma, but in a special extension of the cell called the inner segment (Figure 1B). The inner segment is a region between the nucleus and the outer segment, and it is divided into two regions: the myoid, which contains the endoplasmic reticulum (ER) and Golgi, and the ellipsoid which contains the mitochondria. This places the mitochondria closer to the outer segments, where energy is most needed. In primates, mitochondria within the ellipsoid are heterogenous; those closer to the ER (the apical side) are typically small; and the more distal ones are elongated, line up along the rootlet, and some contact the plasma membrane of the outer segment (a property that is unique to photoreceptors). Mitochondria in the distal region, especially in stressed cones, are atypical; they tend to be swollen and their outer membrane is “expelled” into the space between neighboring outer segments [14]. Cones contain about 10 times more mitochondria than rods, and in some species the size and density of the mitochondria dynamically change between night and day [15,16].

In addition to the ellipsoid, mitochondria are also present in the photoreceptors’ presynaptic terminals, the compartments that contact horizontal and bipolar cells. And of course, all other cells in the retina also contain mitochondria, but to the best of our knowledge, these have a more typical structure and function. As will be detailed in Section 6, several retinal diseases involve mitochondrial dysfunction at an early stage, where VDAC is involved.

### 2.2. Meeting the Energy Demand

Though photoreceptors have numerous mitochondria that can meet the energy demand, it seems that photoreceptors use several pathways for energy production in addition to oxidative phosphorylation (OXPHOS). Several studies showed that, similar to tumor cells, about 80–90% of the energy in rods and cones is obtained by a process referred to as aerobic glycolysis or the Warburg effect [17,18,19,20,21]. Here, we will refer to it as the Warburg effect since aerobic glycolysis may be confusing [22]. An important enzyme in glycolysis is hexokinase (HK) which is located on the outer mitochondrial membrane (OMM) and interacts with VDAC. The common isoform of HK is HK1 and though photoreceptors express HK1, like cancer cells, they express HK2 at a higher concentration [23,24,25,26,27].

Like HK1, HK2 is localized to the OMM where it binds VDAC. As with HK1, this interaction may support VDAC opening and the release of ATP. The released ATP then drives the first step of glycolysis, i.e., phosphorylation of glucose, and this is further processed to produce pyruvate. In many cells, pyruvate enters the mitochondria for the Krebs cycle and OXPHOS, but in dividing cells and in photoreceptors, pyruvate is also converted to lactate outside of the mitochondria—the Warburg effect. The enzymes that facilitate and regulate the Warburg effect are pyruvate kinase M2 (PKM2), lactate dehydrogenase A (LDHA), and hypoxia-inducible factor-1 (HIF-1). PKM2 catalyzes the conversion of phosphoenolpyruvate into pyruvate and is the predominant form expressed in the inner segments of rods and cones. LDHA converts pyruvate into lactate, and the whole process produces glycolytic intermediates that are available for biomass synthesis [20,21,28,29]. HIF-1 is a transcription factor that upregulates glycolysis by increasing the expression of HK2 and inhibiting OXPHOS.

Several studies suggest that additional pathways for energy production occur in the outer segments. By isolating bovine outer segments and testing for ATP and NADPH production, Hsu and Molday suggested that glycolysis and Hexos monophosphate pathways occur in these compartments [30]. In other studies, energy appeared to be produced in the outer segments by extramitochondrial oxidative phosphorylation [31,32]. These studies relied on oximetry and ATP synthesis and cytochrome *c* oxidase (COX) assays performed on purified outer segment discs and purified mitochondria. These pathways probably do not use VDAC, since existing evidence (Section 3) indicates that it is not present in the outer segments.

Finally, a recent paper used organotypic retinal explants and compared the expression of metabolic enzymes between retinal explants with and without RPE [33]. The study suggests that photoreceptors use several pathways including the mini-Krebs cycle, an alanine-generating Cahill cycle, and a lactate-releasing Cori cycle (the Warburg effect) for their energy. Furthermore, the authors of this study claim that rods rely more on oxidative phosphorylation, while cones depend mostly on glycolysis and the Cahill cycle (also called the glucose–alanine cycle where glutamate can be used). Overall, it is clear that photoreceptors use different pathways to produce energy, and this provides photoreceptors with a remarkable versatility and flexibility, allowing them to dynamically adapt the timing and quantitative output of energy metabolism.

### 2.3. Other Functions: Protection and Apoptosis

Functionally, mitochondria in the inner segments of rods and cones not only supply energy but also buffer calcium and protect the cell bodies from the relatively high Ca^2+^ concentration present in the outer segments in the dark [34,35]. In addition, the elongated shape and refractory properties of mitochondria are thought to act as wave guides and microlenses to focus light to the outer segment and increase the probability of photon capture [15,36,37]. Several proteins bound to the mitochondria help to protect the cells, such as HK2. The binding of HK2 to mitochondria can be stimulated by light via AKT-mediated phosphorylation [23]. It has been suggested that phosphorylation by AKT maintains HK2 association with the OMM through its coupling to VDAC, and that this prevents cytochrome C (Cyt C) release and apoptosis. In addition, HK2, when bound to mitochondria, prevents apoptosis by restricting BAX and BAK oligomerization [38] (Figure 2).

Thus, both HK2 and VDAC may work together not only to produce energy, but also to inhibit apoptosis and protect cells [27]. Accordingly, deleting HK2 from rods leads to age-related photoreceptor degeneration. In the absence of HK2, there is an increase in the expression of HSP60 in the outer segments and outer plexiform layer, and an increase in VDAC in the inner segments, the outer plexiform layer, and in the inner nuclear layer [29]. It is therefore reasonable to conclude that HK2 is an important mediator of the Warburg effect in rods, and when this pathway is defective, the retina compensates for it. In this case, compensation may occur by the recruitment of HSP60 to protect the retina. Finally, while mitochondria are designed to maintain healthy cells, they are also involved in processes that lead to cell death.

## 3. VDAC Expression in Healthy Retina

VDAC, a voltage dependent pore, is located in the OMM (see reviews by [39,40]). In addition, VDAC is expressed in the plasma membrane of certain cells [41]. Its best-known function is to support metabolic processes and cell homeostasis by shuttling ADP, ATP, NADH, Ca^2+^, pyruvate, and other metabolites (Figure 2A). It is well known that VDAC has three isoforms and in most cells, the most common one is VDAC1; yet mice lacking VDAC1 are viable [42]. Deleting VDAC2 is lethal, while deleting VDAC3 leads to infertility [43,44]. Therefore, though the three isoforms are similar, their precise functions must be different, and their expression is likely to be tailored to a cell’s environment and function.

With this in mind, it is important to identify the isoforms and their precise localization in different retinal cell types. The first published work to identify VDAC in retinal neurons used an antibody against VDAC (no specific isoform), and as expected, it showed specific staining in mitochondria. This included mitochondria in the inner segments of photoreceptors—mainly in the ellipsoid where mitochondria are present; in the inner nuclear layer—in the cell bodies of horizontal cells, certain bipolar and amacrine cells; and in the ganglion cell layer—in the cell bodies of ganglion cells [45] (Figure 3A). In addition, staining was observed in both plexiform layers where mitochondria are present in neuronal processes. Similarly, staining using anti-VDAC by Rajala et al. [23] was positive in the inner segments, in horizontal cells, and in bipolar cells, but strangely, not in ganglion cells. Staining using another anti-VDAC was strong in most layers (the ganglion cell layer was not in the picture) but weaker in the inner segments [29]. Another attempt to reveal the localization of VDAC1 showed strong staining in the inner segments, OPL, and IPL and faint staining in the somas in the INL [46]. The inconsistent staining patterns indicate that when using immunocytochemistry, most antibodies against VDAC are not sufficiently specific for the VDAC isoforms.

For the purpose of this review, in order to determine the expression pattern of each isoform, we attempted to analyze single-cell transcriptomics data from several groups. These data sets are freely available in Spectacle (a free viewer of single-cell RNA data from retinas; https://singlecell-eye.org/app/spectacle). Data from the different groups indicated that most retinal cells in different species have transcripts for all three isoforms [47,48,49,50,51,52,53,54,55,56,57,58,59,60]. However, we found that the data are very noisy and inconsistent between groups, making it difficult to draw more specific conclusions.

Using in situ hybridization for VDAC1 in the mouse retina, Gincel et al. showed its transcript to be present in most cell bodies, but not in photoreceptors [45] (Figure 3B). These results are consistent with those of another study in which transcripts from specific retinal cell types tested by mRNA amplification showed that VDAC1 mRNA is expressed by most retinal cells but not rods [61]. We further noticed in this data set that most of the retinal cells express more VDAC1 than photoreceptors do. However, for most of these cells, the VDAC1 transcript was not the most common one—most retinal cell types expressed more VDAC2 and VDAC3. Together, these data suggest that VDAC1 is either not expressed or lightly expressed by photoreceptors.

In summary, all three isoforms of VDAC are expressed in the retina. However, the precise expression and localization of the isoforms in the various cell types are unknown due to the lack of specific antibodies for each of them.

## 4. Differential Functions of VDAC Isoforms

As mentioned above, the main function of all VDAC isoforms is to support metabolic processes and cell homeostasis. But VDACs are also involved in cell death processes including apoptosis, necrosis, and ferroptosis. The precise function and the pathway used to execute each function vary depending on the isoform, on the cell type, and on the conditions and environment of the cell. In general, it is now accepted that the functions of the three VDAC isoforms, while similar, are not redundant. It is therefore likely that the nuances in the isoforms are beneficial for the retina, which has a high metabolism and vulnerability to free radicals as stated earlier.

### 4.1. VDAC1

While the precise localization of VDAC1 in the retina is still questionable, its function in retina was investigated by knocking it down using morpholino antisense RNA [46]. In these experiments, VDAC1 expression as measured by Western blots was reduced by about 45%, and this reduction greatly affected retinal structure: the retina was thinner, cells in the nuclear layers were undergoing apoptosis (as labeled by the TUNEL assay), and the number of three tested cell types (photoreceptors, horizontal cells, and ganglion cells, but not rod bipolar cells) was greatly reduced. Thus, VDAC1 seems essential in maintaining the health of most retinal cell classes. Unfortunately, the authors did not validate their immunostaining using their knockdown model; thus, it remains unknown which cell types were directly affected by the reduction in VDAC1 expression. Further support for VDAC1 function in retinal health and disease comes from different disease models as will be discussed in Section 6.

### 4.2. VDAC2

The particular function of VDAC2 in the retina and especially in photoreceptors has not been reported. Here, we will speculate on its function based on studies in other tissues [62] and the unique needs of the retinal cell types presumed to express it. Speculation is difficult as it appears that VDAC2’s interactome and function vary depending on the function of the particular cell type/tissue and on the insult they experience [44,63,64]. Nonetheless, a thought experiment may advance the field.

Regarding the metabolic function, several studies have shown that VDAC1 binds HK2 and helps to tether it to the OMM to protect the cells [65,66,67]. One study showed that VDAC2 also binds HK2 and that this interaction is even more protective than that with VDAC1 [68]. In photoreceptors, VDAC2 is colocalized with HK2 in the inner segments where HK2 is necessary for the Warburg effect. Thus, in photoreceptors, the function of VDAC2 is to support the first step of glycolysis, i.e., to ensure that both OXPHOS and Warburg effect supply the necessary energy.

The protective function of VDAC2 is connected to at least two of its features. First, it is known that hVDAC2 has nine cysteine residues, which are thought to function in sensing reactive oxygen species [69]. It is therefore likely that its relatively high expression in the retina, both in photoreceptors and the inner retina, is protective against the accumulation of ROS. In support of this, it has been shown that when 5 L rat hepatoma cells are exposed to pollutants, the expression of VDAC2 increases and this protects the cells [70].

The second feature has to do with VDAC2’s interaction with HK2. When VDAC is absent, HK2 detaches from the OMM enabling BAX to bind mitochondria and trigger apoptosis [38,67]. Furthermore, VDAC2 also interacts directly with both BAX and BAK [62,71,72,73]. With respect to BAK, VDAC2 inhibits it by reducing its oligomerization: cells deficient in VDAC2, but not those lacking VDAC1, exhibit enhanced BAK oligomerization [44,74] (see schematic illustration in Figure 2). In fact, some studies claim BAK interacts exclusively with VDAC2 [62,71]. With respect to BAX, several studies have shown that the pro-apoptotic function of BAX requires VDAC2 [63,75,76]. Using the human hepatoma HepG2 cell line, it was shown that quinocetone, a toxic chemical, increases cell death when VDAC2 oligomerizes, and DIDS (4′4′-diisothiocyanostilbene-2,2′-disulfonic acid—a blocker of VDAC oligomerization) blocked oligomerization and cell death [77]. These interactions of VDAC2 with BAK and BAX are relevant to photoreceptors since they express both these pro-inflammatory agents which can initiate apoptosis and contribute to photoreceptors’ degeneration under an insult [78]. An overexpression of VDAC2 in rods leads to a thinner ONL layer and reduced retinal light responses by electroretinogram (ERG) recording. This supports the inference that an increase in VDAC2 expression may cause its oligomerization to promote apoptosis, as does overexpressed VDAC1.

### 4.3. VDAC3

If indeed VDAC3 is highly expressed in the retina, it is interesting to speculate about its function. For a long time, the channel properties of VDAC3 were difficult to decipher because it was difficult to isolate this protein and insert it into a lipid bilayer membrane [79]. However, by finding the right conditions, a purified hVDAC3 sample was successfully reconstituted into a planar lipid membrane and it is now established that VDAC3 does form a voltage-sensitive pore with similar properties to those of VDAC1, and that it can support bioenergetic transport [80]. However, VDAC3’s structure is unique, and it must have some non-redundant functions.

A special property of VDAC3 is that it has six cysteine residues, four of which are exposed to the intermembrane space where they are easily accessible to soluble oxidative molecules [69,81]. The role of these residues has been proven convincingly in HAP1 cells that were challenged with oxidative stress (such as applying the complex I inhibitor rotenone). In these cells, deleting VDAC3 caused more ROS toxicity and lower cell viability than deleting VDAC1 [79,82]. Furthermore, substituting the cysteine residues in VDAC3 with alanine removed the protective feature of this pore. That VDAC3 can protect against damaging ROS’s effects was also shown in vivo in mice. Depending on the diet, VDAC3 knockout mice [83] had a higher blood pressure, more ROS accumulation, and more damaged mitochondria than WT on a similar diet. It has also been shown that when cells are exposed to rotenone or other oxidants, they increase their VDAC3 expression; presumably to provide better protection for the cells. Thus, VDAC3 senses reactive oxygen species that might be pouring into the intermembrane space and activates pathways that quench ROS to protect cells. Thus, its expression in the retina likely protects the latter from oxidative damage.

Why would cells in the inner retina express more VDAC2/3 than photoreceptors? As we know, when light enters the eye, it passes through all retinal layers before it is absorbed by the photoreceptors; this likely places these cells at a greater risk of oxidative damage than the photoreceptors themselves. The protective properties of VDAC2 and VDAC3 by different pathways that are discussed above are therefore especially useful for the retina. It would be interesting to test whether VDAC3 expression in the retina increases because of exposure to light, or if it is high by design starting early in development when the eyes are still closed.

### 4.4. VDAC1, 2, 3 in a Retinal Pigment Epithelium Cell Line

To the best of our knowledge, VDAC isoforms in the retinal pigment epithelium are not clearly identified. However, several studies have addressed VDAC localization and function in the RPE1 cell line. In these studies, all three VDAC isoforms were expressed, localizing to their mitochondria and also to the centrosomes and cilia [84,85]. Within the centrosome, the precise localization for the three isoforms differed. Functionally, while VDAC1 and VDAC3 negatively regulated ciliogenesis, VDAC2 appeared to promote the maturation of primary cilia. How the different VDAC isoforms execute their function—whether they function as pores connected with the small membrane fraction that associate with the cilia or in a different configuration—is not known. Also, to what extent one can use these data to infer about the function of VDAC in retinal pigment epithelium in vivo is not clear since although RPE1 was developed from human RPE, the cells do not exhibit several essential characteristics of retinal RPE including polarity and cilia development.

## 5. Mitochondrial State and VDAC Expression with Aging

It is well known that vision declines with age [86] due to changes in retinal health and a reduced number of rods. There are probably many reasons for this, including light exposure and a high demand for energy that lead to ROS accumulation and apoptosis. These conditions lead to profound changes in mitochondria in photoreceptors. ATP production is reduced with age; the mitochondrial ultrastructure changes, probably due to an increase in the expression of the fission (Fis1) and fusion (Opa1) proteins [87,88]. Mitochondrial membrane potentials are reduced significantly, and their membrane permeability increases. The level of fragmented and degraded mitochondrial DNA (mtDNA) increases significantly [89]. Finally, the mitochondrial marker Tom20 declines, suggesting that the number of mitochondria declines with age [90]. Interestingly though, in some studies, the expression and activities of complex I, II, and III did not go down when mitochondria showed other damage, perhaps suggesting a mechanism to compensate for the damage and decline in their number [91].

Regarding VDAC expression, Western blots of the whole retinas in young and old macaques showed that while the expression of Tom20 decreased, the expression of VDAC increased with age [90]. This overexpression of VDAC per mitochondrion might contribute to apoptosis which may explain the reduction in the number of rods in old primates. Support for increased VDAC expression with age comes from our studies where mRNA for VDAC1, 2, and 3 were measured using qRT-PCR during development and at 3 months of age. For VDAC1 and VDAC2, the mRNA expression was relatively stable during development from P10 to P28 and increased at 3 months of age. In contrast, the mRNA expression of VDAC3 remained stable from P10 to 3 months.

## 6. Defective Mitochondria and Altered VDAC Expression in Diseased Retina

It has been shown in multiple diseases such as amyotrophic lateral sclerosis, irritable bowel syndrome, Lupus, Alzheimer’s, myocardial diseases, Type 2 diabetes, and others that when concentrations of ROS increase or the concentration of Ca^2+^ falls below, or rises above, its homeostatic concentration, VDAC1 is overexpressed and this leads to its oligomerization [92,93,94,95]. This in turn leads to the release of cytochrome C from the intermembrane space, the activation of caspases, and apoptotic cell death [96]. Quite amazingly, preventing VDAC1 oligomerization can alleviate the conditions and prolong the viability of the affected cells. With such abundant evidence, it is natural to ask whether a similar intervention can also alleviate retinal diseases leading to an improvement in vision. The first question that should be asked in this context is which retinal diseases are associated with defective mitochondria.

### 6.1. Retinitis Pigmentosa

In Retinitis Pigmentosa, a photoreceptor degenerative disease, vision loss starts with the degeneration of rods followed by that of cones (see reviews [97,98,99,100]). Despite considerable research, the precise chain of molecular events leading to degeneration is still far from understood. There are over 300 genetic mutations that lead to rod degeneration [101]. The major confound in the underlying mechanics of the disease is that the integrity of cones depends on that of rods. The main mouse models for retinitis pigmentosa are rd1 and rd10, where PDE6β, the enzyme that hydrolyses cGMP in rods, is defective. This leads to accumulation of cGMP, followed by the accumulation of Ca^2+^ [102,103] and cell death. While these basic facts are clear, the precise triggers for degeneration are still debated. The difficulty in understanding these processes arises from several reasons. Although certain apoptotic mechanisms are universal, they appear to be different in rods than in most cells. Photoreceptors, being unique in their signal transduction and high metabolic demands, likely use unique pathways with different regulatory mechanisms. For example, it is well established that Ca^2+^ overload triggers apoptosis in most cells, and this is true in photoreceptors as well. However, in photoreceptors, even mutations that reduce intracellular Ca^2+^ concentration cause rod degeneration. In photoreceptors, Ca^2+^ extrusion mechanisms are retained in different compartments [104], and this high level of compartmentalization is crucial for all of their functions. The consequences of these specific needs and adaptations in photoreceptors make it difficult to generalize the function of VDAC in these cells.

Though the precise mechanisms of degeneration still need to be elucidated, it has been shown that already by Postnatal Day 3 of rd1 retinas, the mitochondria in the inner segments are swollen with less inner membrane cristae, and this abnormal development continues, resulting in deformed mitochondria [105]. Existing evidence suggests the accumulation of ROS in these abnormal mitochondria. The application of idebenone, an analogue of CoQ10 that can protect mitochondria, slowed down some aspects of retinal degeneration. In another paper, ROS were targeted using fullerenols as antioxidants [106]. The results were impressive, as fullerenols reversed almost every aspect of the degenerating photoreceptors. Compared to untreated rd1 retina, the treated retina had a thicker outer nuclear layer and healthier photoreceptors expressing more of the essential cascade elements such as rhodopsin, Gnat-1, and arrestin. It also reduced the mtDNA that accumulated in untreated rd1, and most importantly, it greatly improved the ERG a- and b-waves. In connection to VDAC, this study showed that VDAC1 transcript in rd1 was upregulated relative to the control, and fullerenols reduce it almost to its WT level. These results suggest that VDAC1 is involved in cell death in degenerating photoreceptor diseases.

Further support for this interpretation is obtained by experiments on a photoreceptor cell line, 661W. In these experiments, cells that were exposed to H_2_O_2_ accumulated mtDNA and cytochrome C in the cytosol; both were reduced with fullerenols [106]. Since the release of cytochrome C from mitochondria is likely through the VDAC pore [107,108], it is reasonable to conclude that VDAC oligomerizes. Thus, it appears that VDAC1 in rods in the rd1 mouse model is overexpressed resulting in the leakage of mtDNA and cytochrome C followed by the activation of pro-inflammatory pathways and cell death. Though VDAC is very likely involved, its precise role in rd1 needs to be studied to further substantiate this interpretation. More insight will be obtained by examining the transcripts of VDAC2 and VDAC3 in photoreceptors of rd1 at different stages of retinal development.

Examination of the mitochondrial structure in rd mice in the Xu lab showed that it was severely compromised by P12 (before the rods started to degenerate). Not only were the cristae damaged but there were fewer mitochondria compared with age-matched wild type mice. Consistent with the abnormal structure, between P12–P14, mitochondrial function was impaired with increased ROS and decreased ATP production. The transcript (mRNA) levels of all the three isoforms of VDAC were higher at P10–P18 than those in age-matched wild type mice. However, thus far, we have failed to rescue rd10 retina with in vivo application of VBIT-4, VBIT-12, or DIDS. It therefore appears that the overexpression of VDAC may not be the major contributor to the degeneration in rd10 retina. Alternately, the overexpression of VDAC transcripts in rd10 rods might be a compensatory mechanism to try to increase ATP production in these mitochondria.

### 6.2. Age Related Macular Degeneration (AMD)

AMD is a common retinal disease that affects older individuals (see reviews by [109,110]). There is strong experimental evidence from human donors and animal studies that the early stage of the disease is characterized by mitochondrial dysfunction in the retinal pigment epithelium [111,112,113,114,115,116]. A major consequence of the altered metabolism of epithelial cells is that photoreceptor health and function (which is highly dependent on them as they supply glucose, absorb lactate, phagocytose the outer segment during renewal of disks, and recycle 11-Cis retinal) are severely compromised, eventually leading to their death.

The changes in mitochondria include morphological changes, protein modifications, the up- and downregulation of many proteins including those involved in mitochondrial trafficking and apoptosis [113,114], and an increase in levels of fragmented mtDNA [117,118,119]. Though the literature on AMD is prolific, the role of VDAC has not been extensively studied; hence, the contribution of VDAC to RPE degeneration or its protection is not understood. Nordgaard et al. [113] found that VDAC1 expression in different stages of AMD changed with the progression of the disease. However, while their quantitative estimates of VDAC1 expression using MALDI-TOF analysis showed a gradual increase in VDAC1 with disease progression, their quantitative estimates of VDAC1 intensity from Western blots (which did not label all the spots) showed decreased expression, making it difficult to interpret the data. Also, in this analysis, all of the identified spots corresponded to monomers (which might be truncated or phosphorylated). Thus, while this study suggests a change in VDAC1 expression with AMD, further research is needed to determine which VDAC isoform is involved and whether it oligomerizes.

Results from the Shoshan-Barmatz group show that when the RPE cell line ARPE-19 was exposed to NaIO_3_, an in vitro model of AMD, both apoptotic and necrotic cell death increased, as did the expression levels of VDAC1 and its oligomerized forms. Treatment with the VDAC1 oligomerization inhibitor VBIT-4 protected against the NaIO_3_-induced necrosis and apoptosis. Thus, VDAC1 overexpression and oligomerization are linked to AMD.

### 6.3. Glaucoma, Ischemia/Reperfusion, and Mechanical Trauma

Glaucoma is an umbrella of diseases characterized by the progressive degeneration of retinal ganglion cells and the optic nerve head leading to constricted visual fields and eventually total blindness. The predominant cause of the disease is an increase in intraocular pressure (IOP), which in turn increases pressure on ganglion cells’ axons (see reviews by [120,121]). While the exact mechanisms of RGC death in glaucoma are unknown, there is a clear link between mitochondrial dysfunction and selective RGC death. The retinas of late-stage glaucoma patients appear morphologically abnormal with indications of apoptosis, necroptosis, and ferroptosis. Regarding VDAC, using a retinal ischemia/reperfusion model in mice, one study showed a reduction in VDAC1 expression in the retina [122], but this study evaluated only the monomer. Nonetheless, the involvement of VDAC1 oligomerization in glaucoma can be deduced from an experiment using VBIT12 [123], a specific inhibitor of VDAC1 oligomerization. In this study, the authors injected physiological saline to acutely elevate intraocular pressure in the rat retina. With this injury, dying cells were seen in all nuclear layers. VBIT12 pre-treatment reduced the number of cells that were stained by TUNEL or by propidium iodide in both the inner nuclear and the ganglion cell layers. This study also showed that both cleaved capase-3 and phosphorylated MLKL (mixed lineage kinase domain-like protein), which increased under ischemia/reperfusion, were reduced by VBIT12, suggesting that not only do both apoptosis and necroptosis occur in glaucoma–ischemia models, but that both these processes are mediated by VDAC oligomerization. Unfortunately, this study neglected to show VDAC overexpression and oligomerization in the injured retina.

The function of VDAC1 and its oligomerized form was also studied in a model of glaucoma that inflicted mechanical trauma to the eye. As measured by Western blots of the whole retina, mechanical trauma increased VDAC1 levels two hours after the injury was inflicted; interestingly, most of the increase in VDAC1 was localized to the ER and plasma membrane, and this increase lasted for less than 6 h [46]. Nonetheless, the involvement of VDAC1 was clear when DIDS, an inhibitor of VDAC oligomerization, was injected intraocularly. The injection of DIDS reduced the number of TUNEL-, propidium iodide-, and iNOS-labeled cells. In addition, DIDS reversed the expression trends of cytokines after the mechanical lesion: it increased anti-inflammatory cytokines (that were reduced by the lesion) and decreased pro-inflammatory cytokines (that were increased by the lesion). It would be interesting to test whether the plasma membrane and ER localization in this glaucoma model contributed to triggering cell death, perhaps by compromising Ca^2+^ concentration.

Support for the involvement of VDAC oligomerization in injuries caused to RGCs is also obtained from studies that used the retinal precursor cell line R28. This study showed that injuring the cells by oxygen-glucose deprivation and letting them partially recover by oxygen supplementation (OGD/R) leads to cell death by both apoptosis and necroptosis [123]. OGD/R injury caused multiple changes in mitochondria including morphology, reduced membrane potential, increased ROS, increased Ca^2+^ concentration, and increased permeability to cytochrome C. Central to VDAC function in RGC diseases, Wan et al. found that VDAC1 in OGD/R-treated R28 cells is overexpressed and oligomerized. Furthermore, inhibiting oligomerization with VBIT12 greatly improved cells’ survival as seen both by TUNEL and propidium iodide staining. Testing markers for apoptosis and necroptosis showed that cleaved caspase-3, phosphorylated MLKL, LDH, and cytoplasmic cytochrome C were reduced by the treatment, suggesting that VDAC1 oligomerization is involved in both apoptotic and necroptosis.

## 7. Conclusions

The retina expresses all three isoforms of VDAC, but their precise distribution, localization, and function are still unknown. Nonetheless it is clear that all VDAC isoforms contribute to the health of the retina. VDAC1 contributes to most retinal cell types, and VDAC2 likely contributes particularly to the photoreceptors’ integrity and their special metabolic demands. As for VDAC3, it is likely highly expressed in the retina to protect it from ROS generated by its high light exposure and metabolism. It is also clear that the isoforms’ expression changes in several major retinal degenerative diseases and in certain models of glaucoma; blocking VDAC oligomerization prevents apoptosis of RGCs, making this approach an attractive therapeutic target of glaucoma. However, given that many of the studies blocking VDAC oligomerization were conducted in in vitro models of disease, the following questions must be answered in vivo before VDAC can be targeted as a treatment. (1) How does the underlying cause of the disease alter the metabolism of the affected cells? (2) How does the altered metabolism affect VDAC expression, oligomerization, and localization in different cell types? (3) If VDACs are involved in cellular metabolism in retinal neurons as is reasonable to expect, how does their expression correlate with a cell’s energy demand? (4) Does preventing apoptosis retard the progression of the disease and improve/stabilize vision? Answers to these questions will not only better inform us on the practicality of blocking VDACs to treat these diseases but shed invaluable light on their mechanics. Future research into the expression and function of these isoforms will be well worth investing in.

## Figures and Tables

**Figure 1 biomolecules-14-00654-f001:**
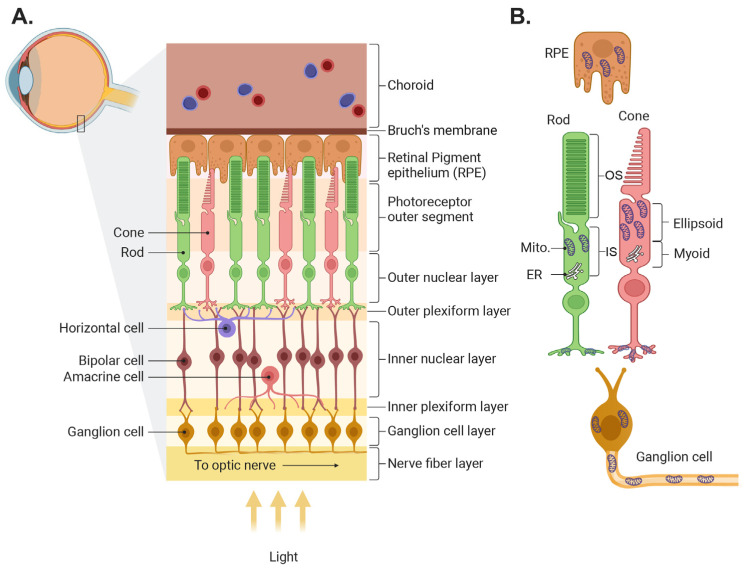
Illustration of retinal structure and localization of its mitochondria. (**A**) Schematic of the retina showing the different layers and the main retinal cell types. (**B**) Mitochondrial location in several retinal cell types. In photoreceptors, mitochondria are located mainly in the ellipsoid in the inner segments and in the axon terminals. Mito., mitochondria; ER, endoplasmic reticulum. The figure was created by Biorender (biorender.com).

**Figure 2 biomolecules-14-00654-f002:**
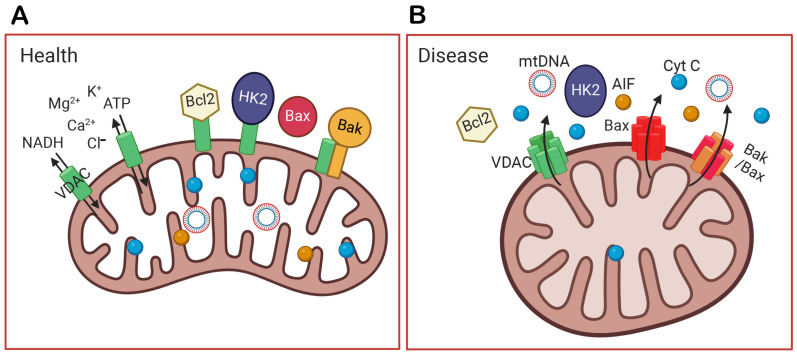
The function of VDAC in healthy and diseased cells. (**A**) In health, VDAC not only transports ions, NADH, and ATP out of mitochondria but also binds Bcl2, HK2, and Bak to prevent apoptosis. (**B**) In a diseased condition, VDAC is overexpressed, oligomerizes, and forms a larger channel. It unbinds HK2, recruits Bax to the outer mitochondrial membrane (OMM), and oligomerizes with itself or with Bax/Bak. The oligomerization of VDAC, Bax, or Bak/Bax form large holes in the OMM, and this allows the release of mitochondrial DNA (mtDNA) and apoptotic factors, like cytochrome C (Cyt C) and the apoptosis-inducing factor (AIF). Their release into the cytoplasm initiates the apoptosis of the cell and inflammation. The figure was created by Biorender (biorender.com).

**Figure 3 biomolecules-14-00654-f003:**
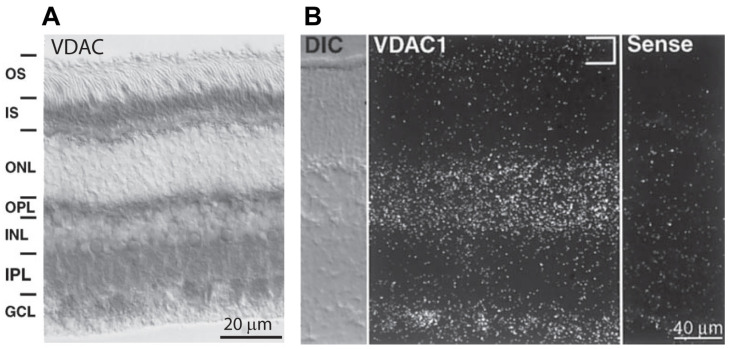
Expression of VDAC isoforms in rabbit and mice retina. (**A**) Immunostaining (using HRP) for VDAC in rabbit retina; strong staining is seen in the inner segments, OPL, IPL, and GCL, and fine staining is seen in the INL around the nuclei of almost all cells. (**B**) In situ hybridization for VDAC1 in a mouse retina section showing hybridization in the INL and GCL, but not in the inner segments ((**A**,**B**) are adapted from Gincel et al. [45]).

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
