# Peer review of "VDAC in Retinal Health and Disease"

_biomolecules, 2024, doi:10.3390/biom14060654_

Round 1

Reviewer 1 Report

Comments and Suggestions for Authors

I found the review by Xu and collaborators very engaging, offering a comprehensive overview of the involvement of mitochondrial channels known as VDACs in both the physiology and pathology of the retina. The manuscript is well structured and provides intriguing insights into the potential of VDACs and related metabolic pathways as new biomarkers, offering promise for hindering the progression of retinal pathologies. This review holds value for both laboratory and clinical researchers. Overall, there are minimal issues that would prevent my recommendation for publication; however, I would suggest some caution in considering VDACs as potential therapeutic targets.

While the review adequaletly discusses the role of VDACs in regulating the metabolic switch between oxidative phosphorylation (OXPHOS) and the Warburg effect (i.e., glycolysis), it could benefit from further emphasizing the significance of this metabolic regulation in the development of degenerative pathologies such as AMD, Glaucoma, and Retinitis Pigmentosa. Much of the evidence reported to support VDACs' involvement in these ocular degenerative pathologies relate to the involvement of VDACs in apoptosis and relies on the use of VDAC blockers like VBIT-4, VBIT-12, or DIDS. However, while VDAC blockers serve as valuable research tools, the practicality of blocking VDAC channels to impede the progression of degenerative pathologies remains dubious due to potential serious side effects.

In addition to these considerations, I have some minor suggestions aimed at improving the readability and quality of the manuscript:

Line 103- Fig1B: It would be beneficial to the reader if the names "inner segment" and "outer segment" were indicated in the picture.

Fig.1: Consider splitting Figure 1 into two separate figures. The first could include the current 1A and 1B, illustrating the structure of the retina and its constituent cells, while the second (1C) could illustrate the function of VDACs in both healthy and diseased cells.

Line 187-188: Instead of stating that “supporting metabolic processes and cell homeostasis by shuttling ADP, ATP, NADH, and Ca2+ is VDAC's primary function," it might be more accurate to describe it as its most well-known function. We cannot definitively assert VDAC’s primary function but just that.. shuttling ADP, ATP, NADH….  was the first to be discovered.

Line 417: Regarding the statement, "Since the release of cytochrome C from mitochondria is through the VDAC pore," I recommend exercising caution, as the mechanism of cytochrome C leakage is not yet fully understood. Presenting alternative hypotheses and including additional references would provide a more exhaustive review.

Overall, the manuscript presents a valuable contribution to the understanding of VDACs' roles in retinal physiology and pathology, and addressing these minor points would enhance its clarity and comprehensiveness.

Reviewer 2 Report

Comments and Suggestions for Authors

The authors in this review bring together crucial information on the retina, which is critical for vision, delving into the metabolic aspect of this tissue and highlighting the importance of mitochondria and mitochondrial porins (VDAC). In particular, they emphasize the different expressions, distributions, and functions of VDAC isoforms in retinal tissue, underlining the pivotal role of VDAC in maintaining the physiological state of the tissue. In addition, they focused on the impact of variation in the expression/function of VDACs and different degenerative diseases in the tissue used for visual function. Overall, the paper is relatively well written. Nevertheless, it could be better organized, and some parts need more attention, as follows:

1.        Paragraph "1. Introduction" (lines 32-37) is too concise; therefore, I would suggest the authors increase this section or incorporate it into paragraph "2 Retinal structure and function" and organize it better.

2.        In the 'Retinal Structure and Function' section (lines 39-98), the authors provide a comprehensive description of retinal tissue's functions and structure. However, proper bibliographic citations, especially in the initial part, are crucial to enhance the paper's academic rigor. I suggest adding the correct citations to ensure the reader can verify the information.

3.        While the authors have cited extramitochondrial oxidative phosphorylation on outer segments, it seems that evidences supporting extramitochondrial electron transport chain complexes are somewhat weak.

4.        The authors discuss the function of VDAC isoforms in the retina, but due to the lack of studies on VDAC2/3, they speculate about their role. It might be beneficial to condense the description of the three isoforms into a single paragraph, considering the level of speculation regarding VDAC2/3. There are indeed missing quotations. In particular I suggest to add the review about VDAC isoforms by Messina et al (BBA 2012), the seminal review by Varda Shoshan Barmatz et al (Mol Aspect Medecine 2010). I also suggest to the authors to take inspiration from the papers about plasma membrane VDAC, which could be of interest in this specialized tissue.

5.        References are missing for lines 381-386.

6.        Regarding line 420, it might be considered risky to speculate about changes in pore size. The study's data suggests an involvement of VDAC in the inflammatory pathway in retinitis pigmentosa.

7.        Line 457 is written in a different font dimension.

8.        Similar to the “Introduction”, the conclusions are too concise. Often, the authors conclude the various subsections in the manuscript with comments/speculations that could be better discussed in the conclusion section. Again, I recommend reorganizing these sections.

Comments on the Quality of English Language

 There si the need of a moderate editing of English language
